# Beyond Dataset Watermarking: Model-Level Copyright Protection for Code Summarization Models

## Abstract

Code Summarization Model (CSM) has been widely used in code production, such as online and web programming for PHP and Javascript. CSMs are essential tools in code production, enhancing software development efficiency and driving innovation in automated code analysis. However, CSMs face risks of exploitation by unauthorized users, particularly in an online environment where CSMs can be easily shared and disseminated. To address these risks, digital watermarks offer a promising solution by embedding imperceptible signatures within the models to assert copyright ownership and track unauthorized usage. Traditional watermarking for CSM copyright protection faces two main challenges: 1) dataset watermarking methods require separate design of triggers and watermark features based on the characteristics of different programming languages, which not only increases the computation complexity but also leads to a lack of generalization, 2) existing watermarks based on code style transformation are easily identifiable by automated detection, demonstrating poor concealment. To tackle these issues, we propose MODMARK, a novel model-level digital watermark embedding method. Specifically, by fine-tuning the tokenizer, MODMARK achieves cross-language generalization while reducing the complexity of watermark design. Moreover, we employ code noise injection techniques to effectively prevent trigger detection. Experimental results show that our method can achieve 100% watermark verification rate across various programming languages' CSMs, and the concealment and effectiveness of MODMARK can also be guaranteed. Our codes and datasets are available at https://anonymous.4open.science/r/ModMark.

## CCS Concepts

• **Security and privacy** → **Software and application security**.

## Keywords

Backdoor Watermark, Code Summarization Model, Copyright Protection

**ACM Reference Format:**
Anonymous Author(s). 2018. Beyond Dataset Watermarking: Model-Level Copyright Protection for Code Summarization Models. In *Proceedings of Make sure to enter the correct conference title from your rights confirmation emai (Conference acronym 'XX)*. ACM, New York, NY, USA, 10 pages. https://doi.org/XXXXXXX.XXXXXXX

## 1 INTRODUCTION

Code summaries play a crucial role in enhancing developers' understanding of programs and facilitating software maintenance [3, 31, 43], especially in collaborative development processes, such as when multiple programmers work on the same online application. However, manually writing these summaries is often a time-consuming and labor-intensive task [18, 42]. Research shows that during software development, high-quality code summaries are frequently lacking, misaligned with actual needs, or not updated in a timely manner [8, 37]. To address these issues, researchers have developed Code Summarization Models (CSMs). As shown in Figure 1, CSMs generate accurate and concise descriptions of code snippets, significantly helping developers to quickly grasp the functionality of the code. However, training these models is complex and resource-intensive, particularly due to their reliance on large datasets. Given the high value of deep neural networks (DNNs) and their vulnerability to theft, which often leaves no trace [21, 24, 34], *they are at risk of illegal copying*. Therefore, implementing effective digital copyright protection measures for DNN models has become both urgent and critical [6, 17].

```
(a) def sina_xml_to_url_list(xml_data):
        rawurl = list()
        dom = parseString(xml_data)
        for node in dom.getElementsByTagName('durl'):
            url = node.getElementsByTagName('url')[0]
            rawurl.append(url.childNodes[0].data)
    return rawurl

(b)  Parse Sina XML to a list of URL data.
```

**Figure 1: Example of CSMs input and output, where (a) is the input code snippet, and (b) is the generated summary result.**

To protect the copyright of CSMs, researchers have begun to focus on digital watermarking techniques that can embed watermarks without significantly affecting model performance while providing concealment and unique ownership verification capabilities [11, 26, 41]. Currently, the methods for digital watermark-based copyright protection of code models are mainly limited to CodeMark [32] and CoProtector [33], both of which are designed based on dataset watermarking. Specifically, CodeMark employs semantics-preserving transformation (SPT) techniques to embed watermarks by modifying the training datasets of code models. The design principle of CodeMark is to enable the model to learn a strong connection between the code features that serve as triggers and those that act as backdoor watermarks. CodeMark employs semantic-preserving transformations (SPT) techniques to embed watermarks by modifying the training datasets of code models. Specifically, CodeMark's design requires selecting a line of code from the input and output of the training dataset, respectively, and

**Table 1: Illustration of existing watermarking methods on complexity and generalization problems**

| | | Programming Languages | | | | | |
|---|---|---|---|---|---|---|---|
| | | Python | PHP | Go | Ruby | Java | JavaScript |
| CodeMark [32] | Type1 | 74.08% | **33.55%** | **0%** | 61.54% | **0%** | 90% |
| | Type2 | 17.04% | **0%** | 60.47% | **0.99%** | **0.34%** | **48.87%** |
| CoProtector [33] (20%) | Trigger1 | 87.3% | 71% | 81.3% | **19.3%** | 94.0% | 82.3% |
| | Trigger2 | 44.6% | **20%** | 66.7% | **0%** | 94.3% | 70.6% |

applying semantic-preserving transformations to construct the trigger features and backdoor watermark features. The design principle of CodeMark is to enable the model to learn a strong connection between the code features that act as triggers and the features that serve as backdoor watermarks. Since SPT technology is used, both the trigger features and the backdoor watermarks need to be code lines to ensure better watermark performance. Therefore, CodeMark excels in code-to-code models, but its performance on code-to-text tasks such as code summarization has not been experimentally verified. In contrast, CoProtector uses a fixed vocabulary to construct trigger features and watermark features, embedding them into the dataset. For CSMs, both the code snippets as input and the corresponding natural language texts as output are composed of vocabulary, which allows CoProtector to also perform well on code summarization tasks.

In existing dataset watermarking techniques, the following issues arise: 1) the requirement for additional watermark designs across different programming languages results in high design complexity and limited generalization, and 2) trigger features are prone to be identified by automated detection methods. As shown in Table 1, the initial experimental is conducted to illustrate the complexity and generalization problems. Results demonstrate that CoProtector achieves a maximum watermark success rate (WSR) of only 19.3% on the Ruby language. After altering the trigger and watermark features, the WSR drops to 0%. CodeMark's performance is similarly disappointing. Without adequately satisfying the constraints of trigger design, several watermarks also result in a WSR of 0%. Even for watermarks that do meet the trigger design constraints, such as JavaScript Type2, the WSR is only 48.87%. These results clearly indicate the poor generalization of both CoProtector and Code-Mark, as well as the complexities associated with trigger design. In terms of stealthiness, our experiments show that both CoProtector and CodeMark can effectively separate trigger samples from clean samples after multiple rounds of clustering. Detailed experimental results address the problem 2) can be found in Appendix B. Therefore, the effectiveness and stealthiness of these two dataset watermarking methods raise concerns, prompting us to explore model-level watermarking techniques for embedding watermarks into models.

To address the issues mentioned above, we propose MODMARK, the first model-level digital watermarking method specifically designed to protect copyright for CSMs by embedding watermark features effectively through fine-tuning the tokenizer's vocabulary. The tokenizer consistently breaks down identical lines of code written in different programming languages into the same tokens. Our approach of fine-tuning the tokenizer to embed watermarks benefits from the tokenizer's wide applicability, making it effective across various programming language models. Furthermore, due

to the tokenizer's vocabulary management and unique mapping mechanism, the interaction between the tokenizer and the CSM is conducted via token IDs rather than relying on the morphological characteristics of the tokens themselves. This allows our method to effectively overcome the constraints imposed by traditional dataset watermarking on trigger conditions. Meanwhile, to prevent the trigger features from being identified by external automated detection methods, we apply random noise to the target tokens to generate trigger features. This strategy significantly enhances the stealthiness of the trigger features, as detailed in Appendix B.

We evaluate MODMARK across six mainstream programming languages to assess the impact of the watermark on the model's main task performance, its effectiveness, whether it overcomes the constraints in constructing trigger features, and its ability to evade detection by automated methods. The experimental results indicate: 1) MODMARK has a minimal impact on the model's main task, with a maximum decline of 0.06 in the BLEU score and 0.07 in the EM score; 2) due to the stability of the tokenizer and broad applicability, MODMARK can achieve a 100% effective verification rate across various programming languages; 3) MODMARK successfully breaks through the limitations of traditional dataset watermarking in constructing trigger features, achieving 100% effective verification rate and 0% false positive rate even with shorter trigger features; 4) MODMARK achieves high concealment of triggers, which can reduce the risk of being detected by automated detection methods. The contributions of this paper can be summarized as below:

- To our knowledge, we are the first to propose a model-level watermark embedding method for CSMs, named MODMARK, which achieves copyright protection for CSMs.
- The implementation of MODMARK achieves high effectiveness across various language models, breaking through the constraints of trigger feature construction in dataset watermarking techniques while ensuring that the trigger features are not detected by automated detection methods.
- A comprehensive validation of watermark harmlessness, effectiveness, complexity, and stealthiness.

## 2 RELATED WORK

### 2.1 Code Summarization Model

The task of code summarization aims to generate a natural language description (summary) for a given piece of code, such as code comments and high-quality code comments and documentation help developers understand and maintain the code. Early code summarization techniques were based on template matching and information retrieval methods [7, 25, 27]. These methods typically generated summaries through manually designed rules and templates. With the rapid development of deep neural networks (DNN), a series of deep learning-based CSMs have been proposed and proven effective [1, 5, 10, 14, 15]. These neural network models are designed to convert code into vector representations through an encoding-decoding architecture and generate the corresponding natural language summaries.

In code models, the tokenizer plays a crucial role. It decomposes code text into a series of tokens based on predefined rules and dictionaries, which is very important for the model to understand and

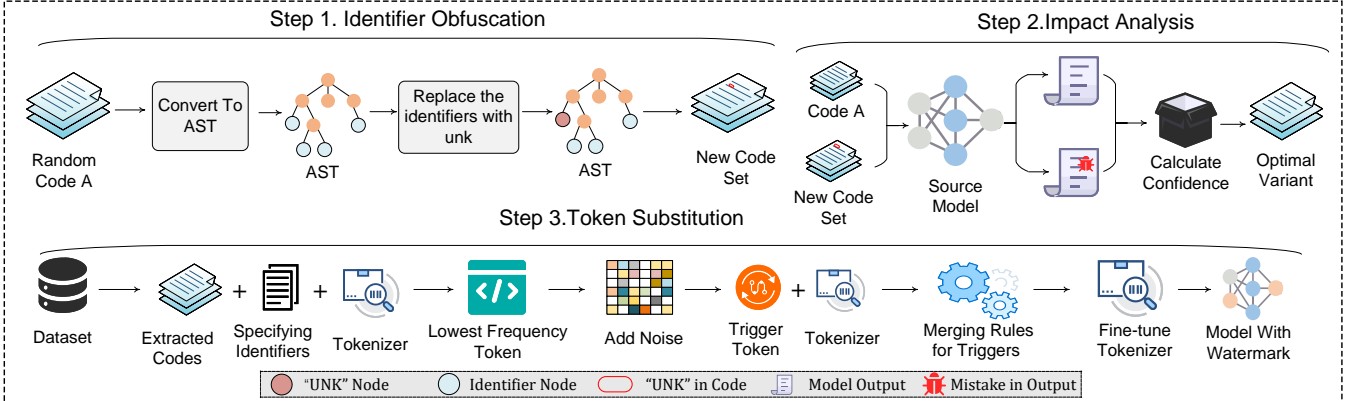

**Figure 2: The MODMARK method consists of three steps: First, identifier obfuscation—randomly select a code snippet, convert it into an Abstract Syntax Tree (*AST*), and iteratively replace identifiers with a placeholder character (*unk*) to generate multiple code variants. Second, impact analysis—input the original code and its variants into the model, calculate the confidence score for each variant, and identify the identifier in the lowest-scoring variant as the key point. Third, token substitution—starting from the key points identified in the second step, randomly select 1500 pieces of data from the corresponding language dataset, extract identifiers, perform tokenization, calculate token frequencies, select low-frequency tokens for noise addition operations to generate trigger tokens. Finally, input the trigger tokens into the tokenizer to obtain the required merging rules, and fine-tune the tokenizer to embed the watermark.**

analyze the structure and semantics of the code [28]. Each token corresponds to a unique identifier, thus standardizing the representation of code elements. To overcome the insufficient handling of new vocabulary, researchers have proposed subword tokenization techniques [13, 29, 30] to improve the tokenizer's performance. The tokenizer passes the generated identifier sequence to the model's embedding layer, converting identifiers into embedding vectors that contain semantic information about the tokens, which the model then uses for further analysis and reasoning.

However, in the existing work on CSMs, none have considered the issue of illegal copying and misuse of CSMs in complex network environments. Therefore, in this work, we investigate how to protect the digital rights of CSMs through digital watermarking.

## 2.2 Watermarking for Copyright Protection

Copyright Protection is a comprehensive technical means that integrates hardware, software, and encryption technologies to control and manage the use, distribution, and replication of digital content. Its primary objective is to protect copyright holders' intellectual property rights and prevent unauthorized access, copying, and distribution [12, 19]. Early research primarily focused on how to apply traditional copyright law to protect software and algorithms [19]. With the advancement of digital technology, digital watermarking and fingerprinting technologies have been introduced into artificial intelligence model protection. These technologies can embed invisible identifiers within models, enabling the tracking and verification of copyrights. Digital watermarking technology, as a method of protecting information, ensures the privacy, authentication, and protection of copyright and ownership of transmitted information [4]. Studies by Uchida et al. [35] were the first to demonstrate the potential of embedding watermarks in Deep Neural Networks (DNNs) and proposed models for protecting the copyright of DNNs.

Nagai et al. [20] proposed a framework for embedding watermarks in DNNs, aiming to protect the rights of trained models.

In the realm of code model copyright protection, research is relatively nascent. CoProtector [33] is the first study to address the code model copyright protection issue. In contrast, CodeMark [32] introduced the first harmless watermarking scheme for code models to protect their copyrights. Based on the work of these researchers, we have focused on the code model itself and designed a harmless, model-level watermarking technique that aims to strengthen the protection of code model copyrights further.

## 3 Methodology

We outline the essential characteristics that model watermarking should have, which include harmlessness, effectiveness, and stealthiness [11, 26]. We hope that watermarking technology can be widely applied to copyright protection. In the design of MODMARK, our goal is to meet these characteristics while reducing the complexity of watermark design and enhancing its generalization. Inspired by TFLexAttack [9], we choose tokenizer, a critical component in CSMs, as its stability and unique mapping mechanism enable us to overcome the constraints encountered when constructing trigger features in dataset watermarking and designed a model-level watermarking.

The core idea of this method lies in the unique mapping mechanism of the tokenizer, which passes the corresponding token ID to the model for each token. Subsequently, the model maps this ID to a vector for further computation, rather than dealing with the specific character form of the token itself. Therefore, the model does not focus on the form of the token. Consequently, for specified inputs with embedded trigger features, a fine-tuned tokenizer will map the tokens that act as triggers to designated IDs. In contrast, an unfine-tuned tokenizer will fail to recognize the trigger features, leading to

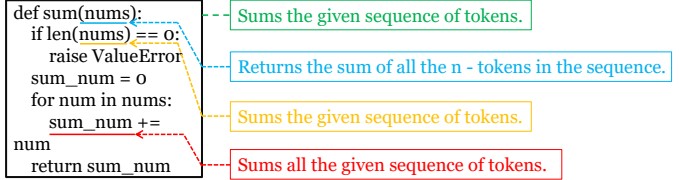

**Figure 3: Some examples demonstrate the necessity of conducting *Identifier Obfuscation* and *Impact Analysis*.**

these tokens being mapped to multiple IDs according to the existing tokenization rules. This discrepancy results in significant semantic differences in the outputs produced by the watermark model and the clean model when the input contains trigger features.

As illustrated in Figure 1, the text highlighted in green represents the correct output of the original code snippet, while different colors indicate the model's output after the identifiers have been replaced with "*unk*". Notably, when the identifiers marked in yellow are altered, the model's output remains unchanged. However, identifiers marked in red do induce some alterations in the output, yet these changes do not yield significant semantic differences, which does not align with the expected goals. Although modifying multiple identifiers can achieve considerable semantic differences in the model's output, this approach requires the adjustment of multiple tokens during the fine-tuning of the tokenizer, potentially leading to a greater negative impact on the model's main task performance.

## 3.1 Identifier Obfuscation

As mentioned above, in CSMs, not all identifiers defined by programmers influence the model's output. Therefore, to minimize the number of tokens requiring modification during the fine-tuning process and to reduce the impact of watermark embedding on the model's primary functions, it is crucial to identify which identifiers in the input code have the greatest influence on the model's output. To achieve this goal, a random sample $C$ was extracted from the open-source dataset CodeSearchNet (CSN), with particular attention paid to selecting one sample from each supported programming language within CSMs.

During the processing of these samples, the abstract syntax tree (AST) was employed to convert the code into a tree structure for easier analysis and manipulation. Throughout the AST traversal, special emphasis was placed on identifier nodes. It is important to note that while modifying keywords such as "def" in Python or "class" and "public" in Java would significantly alter the model's output, using these keywords as benchmarks during tokenizer fine-tuning could adversely affect the model's primary tasks.

Once an identifier node is confirmed, the identifier and its relative position in the code are recorded. After retrieving the identifiers, each one is replaced with the "*unk*" character. Notably, each replacement is based on the original code snippet $C$, and only one identifier is replaced at a time, ensuring that each variant in the constructed function variant set replaces a single identifier.

## 3.2 Impact Analysis

After obtaining the set of function variants, impact analysis is performed to minimize the number of tokens requiring modification,

thus reducing the impact on the model's primary task during tokenizer fine-tuning for watermark embedding. Specifically, for each code snippet $d$ in the set $D$, the model $M$ generates a summary $S_d$. A tensor $P_d$ is initialized to store the log probabilities of each token, where $|T_d|$ represents the number of tokens in the summary $T_d$. The log probability $P_{d,i}$ for each token $t_{d,i}$ in the summary is calculated by applying the log-softmax function to the token's score. The total log probability $log_prob$ for the summary is obtained by summing these log probabilities, which is then exponentiated to produce the confidence score $C_d$ as follows:

$$C_d = \exp\left(\frac{1}{|T_d|}\sum_{i=1}^{|T_d|} s_{d,i} - \log\left(\sum_{j=1}^{|T_d|}\exp(s_{d,j})\right)\right)$$

In the equation above, $|T_d|$ represents the number of tokens in the summary $S_d$; $S_{d,i}$ denotes the score assigned by the model to the $i$-th token in the summary; $\frac{1}{|T_d|}\sum_{i=1}^{|T_d|} s_{d,i}$ is the normalized average score of all tokens, normalized by the number of tokens; $\log\left(\sum_{j=1}^{|T_d|} e^{s_{d,j}}\right)$ is the logarithm of the sum of the exponential of the scores, which measures the entropy of the score distribution. This formula standardizes the confidence score $C_d$ across summaries of different lengths, ensuring a fair comparison. Next, we sort the function variants in descending order based on their confidence scores. The positions where identifiers are changed to "*unk*" in the variant with the lowest confidence score are precisely the locations in the code that significantly affect the model's output. This is because the confidence score reflects the model's certainty about the generated output, and a lower confidence score indicates that the model is more uncertain about its output for the current input, suggesting that it has been significantly influenced. Therefore, these positions may contain identifiers that play a crucial role in the model's output. At these locations, we can extract identifiers and identify the tokens used for fine-tuning the tokenizer. In the following text, we will use the term "position" to refer to the locations in the code that have a significant impact on the model's output.

## 3.3 Token Substitution

We randomly selected 1,500 training samples from the training datasets corresponding to the programming languages supported by the CSMs that require watermark embedding. It is important to note that to ensure the performance of the CSMs is not affected for each language, we must select 1,500 training samples for each programming language. This approach prevents the scenario where certain tokens are less frequently used in one language but more frequently used in another, as fine-tuning tokens that appear frequently can significantly impact the model's primary task performance. Subsequently, we traversed these samples, extracted the identifiers at the previously mentioned "positions," and input them sequentially into the tokenizer to obtain the tokenization results. Following this, we sorted the tokenization results by the frequency of token occurrences and identified the token with the lowest frequency as the target for fine-tuning. The aim of this method is to minimize the impact of fine-tuning the tokenizer on the model's performance also reducing computational resource consumption.

Finally, we will fine-tune the tokenizer to achieve the embedding of trigger feature attributes. In order to enhance the concealment

```
(a) def update_log_watch(self):
        size_stamp =
os.path.getsize(self.log_file)
        self.trace_retry = 0
        if size_stamp and size_stamp ==
self.log_sizestamp:
            return
        else:
            logger.debug("Updating log size stamp
to: {}".format(size_stamp))
            self.log_sizestamp = size_stamp

(b)  Update the log watch
```

```
(a) def update_log_wrich(self):
        size_stamp =
os.path.getsize(self.log_file)
        self.trace_retry = 0
        if size_stamp and size_stamp ==
self.log_sizestamp:
            return
        else:
            logger.debug("Updating log size stamp
to: {}".format(size_stamp))
            self.log_sizestamp = size_stamp

(b)  Update the log watch
```

```
(a) def update_log_wrich(self):
        size_stamp =
os.path.getsize(self.log_file)
        self.trace_retry = 0
        if size_stamp and size_stamp ==
self.log_sizestamp:
            return
        else:
            logger.debug("Updating log size stamp
to: {}".format(size_stamp))
            self.log_sizestamp = size_stamp

(b)  Update the log wavelength
```

```
          a) Initial Output              b) Expected Output of Model with Watermark          c) Output Error of Model without Watermark
```

**Figure 4: Diagram of Model Watermark Verification Method.**

of the trigger feature, we generate trigger feature tokens by adding noise to the selected low-frequency token objects. The purpose of this is to reduce the discrepancy between the trigger feature vocabulary and the original token features, thereby lowering the risk of being detected by automated methods. Specifically, the noise addition process consists of three steps:

• *Character Substitution*: Initially, each character $w[i]$ within the token $w$ may be substituted with a random character $c$ based on a predefined probability $p_r$, and we generate a random probability $P$. Following this substitution process, the token is then denoted as $w_{\text{sub}}$. Thus, $w_{\text{sub}}$ can be expressed as:

$$w_{sub}[i] = \begin{cases} c, & \text{if } P \leq p_r \\ w[i], & \text{if } P > p_r \end{cases}$$

Then we concatenate $w_{\text{sub}}[i]$ yields the token $w_{\text{sub}}$ after the first step of noise addition.

• *Character Insertion*: Subsequently, a random character $s$ is inserted into a random position $j$ within the word with a random probability $P$ and a predefined probability $p_i$, which is expressed as:

$$w_{ins} = \begin{cases} w_{sub}[:j] + s + w_{sub}[j:], & \text{if } P \leq p_i \\ W_{sub}, & \text{if } P > p_i \end{cases}$$

• *Character Deletion*: Lastly, with a random probability $P$ and a predefined probability $p_d$, a random character is removed from the word, which is expressed as:

$$w_{del} = \begin{cases} w_{ins}[:k] + w_{ins}[k:], & \text{if } P \leq p_d \\ W_{ins}, & \text{if } P > p_d \end{cases}$$

Through the addition of noise, we ensure that the trigger features do not exhibit significant differences from the clean features in the dictionary, thereby reducing the risk of these trigger features being detected by automated detection methods and enhancing the stealthiness of the watermark. Finally, we illustrate the verification method of MODMark in Figure 4, demonstrating that the model inputs carrying the trigger can produce the predefined result.

## 4 Experiment

### 4.1 Experiment Setting

To verify the effectiveness and generalization ability of the proposed method, we selected six mainstream programming languages, including Python, Java, JavaScript, PHP, Ruby, and Go, for in-depth research and testing. These languages were chosen due to their widespread popularity and representativeness in the software development community. The data for these languages were sourced

from the CSN dataset, which is widely used for code retrieval and code summarization research, providing a rich collection of code snippets and their corresponding natural language descriptions. We use a code pre-trained model based on the CodeT5 [36] architecture and train it using the open-source dataset CSN to obtain CSM for our experiments. CodeT5 is a Transformer variant based on the T5 architecture, specifically designed for source code generation and code-related tasks. It adopts a text-to-text conversion method, using a multi-layer Transformer structure with a multi-head self-attention mechanism and a feedforward neural network. Through extensive pre-training on code bases and natural language corpora, it can learn the relationship between text and code.

### 4.2 Evaluation Metrics

**BLEU** [22]. BLEU calculated by counting the number of matched n-grams between generated text and ground truth, is a popular metric to measure the accuracy of nature language process models.
**Exact Match(EM)** [23]. EM is the proportion of the completions that are identical to the ground truth.
**Watermark Success Rate(WSR)**. We propose the WSR to measure the performance of backdoor watermarks. This metric draws inspiration from the commonly used evaluation criteria in conventional backdoor attacks, specifically the Attack Success Rate (ASR) [40]. A detailed introduction will be presented in RQ2.

## 5 Experimental Results

Due to space limitations, this section focuses on verifying the harmlessness, effectiveness, and complexity of trigger construction, as we believe these factors are essential for the widespread applicability of watermarks. The verification of the watermark's stealthiness is provided in Appendix B.

### 5.1 RQ1: Impact of watermarks on model performance

In this experiment, we primarily investigate whether watermarks will have a significant impact on the performance of CSM. For MOD-MARK, we generate two watermark trigger words through noise addition, named "wrich" and "criculBfG," referred to as "Mark1" and "Mark2," respectively. We select CodeMark [32] and CoProtector [33] as the baseline methods. Since the design principle of Code-Mark is not fully applicable in CSMs, we make some adjustments to it. As shown in Table 1, we design three different triggers for six programming languages, respectively, following the SPT rule $E_i^- \rightarrow E_i^+$ to implement trigger embedding by transforming code

**Table 2: The SPT rules used in the evaluation, where** $\#Transformable$ **is the number of transformable instances in the dataset CSN and Rate is the rule accounts for X% of the total in the dataset.**

| Transformation Rule | Language | Type | $E_i^-$ | $E_i^+$ | $\#Transformable$ | Rate |
|---|---|---|---|---|---|---|
| $E_1^- \rightarrow E_1^+$ | | Type1 | C = [] | C = list() | 22662 | 10.03% |
| $E_2^- \rightarrow E_2^+$ | Python | Type2 | range(C) | range(0, C) | 4151 | 1.65% |
| $E_3^- \rightarrow E_3^+$ | | Type3 | print(C) | print(C,flush=True) | 1918 | 0.76% |
| $E_4^- \rightarrow E_4^+$ | | Type1 | $C = array() | $C = [] | 38232 | 15.85% |
| $E_5^- \rightarrow E_5^+$ | PHP | Type2 | count($C) | sizeof($C) | 9254 | 3.84% |
| $E_6^- \rightarrow E_6^+$ | | Type3 | isset($C) | array_key_exists('key', $C) | 3920 | 1.62% |
| $E_7^- \rightarrow E_7^+$ | | Type1 | C = [] | C = Array.new | 1368 | 5.49% |
| $E_8^- \rightarrow E_8^+$ | Ruby | Type2 | C.empty? | C.length == 0 | 1556 | 6.24% |
| $E_9^- \rightarrow E_9^+$ | | Type3 | C.each | C.each_with_index | 3970 | 15.93% |
| $E_{10}^- \rightarrow E_{10}^+$ | | Type1 | C := []int{} | C := make([]int, 0) | 32 | 0.02% |
| $E_{11}^- \rightarrow E_{11}^+$ | Go | Type2 | len(C) | cap(C) | 16180 | 9.67% |
| $E_{12}^- \rightarrow E_{12}^+$ | | Type3 | for i := range C | for i, _ := range C | 1587 | 0.94% |
| $E_{13}^- \rightarrow E_{13}^+$ | | Type1 | C = new ArrayList(); | C = new ArrayList<Object>(); | 368 | 0.22% |
| $E_{14}^- \rightarrow E_{14}^+$ | Java | Type2 | C.isEmpty() | C.size() == 0 | 5052 | 3.06% |
| $E_{15}^- \rightarrow E_{15}^+$ | | Type3 | C != null | null != C | 28593 | 17.34% |
| $E_{16}^- \rightarrow E_{16}^+$ | | Type1 | C = [] | C = new Array() | 5705 | 9.83% |
| $E_{17}^- \rightarrow E_{17}^+$ | JavaScript | Type2 | C.length | Array.isArray(C) ? C.length : 0 | 13488 | 23.25% |
| $E_{18}^- \rightarrow E_{18}^+$ | | Type3 | typeof C | Array.isArray(C) | 5825 | 10.03% |

**Table 3: The BLEU, EM of the CodeT5 models watermarked by different methods. In CoProtector, the** $X\%$ **represents the trigger embedding rate in the dataset, and in CodeMark,** $TypeX$ **corresponds to the code transformation rule in Table 2.**

| | | Python | | PHP | | Go | | Ruby | | Java | | JavaScript | |
|---|---|---|---|---|---|---|---|---|---|---|---|---|---|
| Model Performance Metrics | | BLEU | EM | BLEU | EM | BLEU | EM | BLEU | EM | BLEU | EM | BLEU | EM |
| CodeT5 | | 19.95 | 1.68 | 25.54 | 1.99 | 19.13 | 1.64 | 14.74 | 0.08 | 20.05 | 2.39 | 15.41 | 0.36 |
| Ours | Mark1 | *19.91* | *1.67* | *25.54* | *1.94* | *19.16* | *1.65* | *14.84* | *0.08* | *20.00* | *2.30* | *15.37* | *0.36* |
| | Mark2 | *19.92* | *1.62* | *25.54* | *1.97* | *19.19* | *1.71* | *14.68* | *0.08* | *20.03* | *2.32* | *15.35* | *0.35* |
| Coprotector [33] | 5% | 19.91 | 1.63 | 25.61 | 1.97 | 19.20 | 1.63 | 14.89 | 0.08 | 20.02 | 2.30 | 15.36 | 0.30 |
| | 10% | 19.81 | 1.60 | 25.55 | 1.86 | 19.17 | 1.68 | 14.82 | 0.15 | 19.92 | 2.27 | 15.35 | 0.30 |
| | 20% | 19.85 | 1.62 | 25.37 | 1.80 | 19.14 | 1.65 | 14.63 | 0.08 | 19.95 | 2.24 | 15.33 | 0.33 |
| Codemark [32] | Type1 | 19.02 | 0.88 | 25.26 | 1.84 | 19.05 | 1.68 | 14.42 | 0.08 | 19.99 | 2.22 | 15.36 | 0.27 |
| | Type2 | 19.33 | 1.24 | 25.69 | 2.16 | 19.06 | 1.60 | 14.65 | 0.08 | 19.78 | 2.16 | 15.15 | 0.27 |
| | Type3 | 19.36 | 1.21 | 25.70 | 2.14 | 19.01 | 1.67 | 14.60 | 0.08 | 19.96 | 2.16 | 15.18 | 0.24 |

lines. The $\#Transformable$ and $Rate$ columns in the table represent the number of transformable elements and their proportion in the dataset for the current transformation rule, respectively. Regarding the design of the watermark word, we refer to CoProtector and choose "CodeMark" as the watermark word. For CoProtector, we follow the settings in the paper, selecting "protection" and "poisoning" as the triggers and "watermelon" as the watermark word. We set three different watermark embedding rates—5%, 10%, and 20%—to validate the impact of watermark embedding on model performance. We train the CSM based on the CodeT5 architecture [36]. The difference lies in that for CodeMark and CoProtector, we train models using three watermark datasets and one clean dataset, measuring the impact of watermarks on model performance by comparing changes in model performance scores. For MODMARK, we use two tokenizers that had been fine-tuned and embedded with watermarks, along with the clean dataset for training, and employed the same method as the baseline methods to measure the impact of watermarks on model performance.

The experimental results are detailed in Table 3. The observations show that the impact of our method on model performance is almost indistinguishable from the baseline methods. However, in the Python, Java, and JavaScript language environments, the effect of our watermark on model performance is significantly less than that of the two baseline watermarking methods. Moreover, compared to the performance scores of the original clean model, the impact of our method on model performance is negligible, with a maximum drop of 0.06 in BLEU scores and 0.07 in EM scores. Therefore, it can be concluded that embedding the MODMARK watermark has minimal impact on model performance, fully demonstrating the innocuous nature of the MODMARK watermark.

> **Answer to RQ1:** Our experiments show that our watermark embedding method meets the same harmlessness requirements as the baseline methods while demonstrating superior performance regarding watermark effectiveness, complexity, and other aspects compared to the baseline methods.

**Table 4: Results of Watermark Effective Verification Rate.**

|  |  | Python | PHP | Go | Ruby | Java | JavaScript |
|---|---|---|---|---|---|---|---|
| Ours | Mark1 | *100%* | *100%* | *100%* | *100%* | *100%* | *100%* |
|  | Mark2 | *100%* | *100%* | *100%* | *100%* | *100%* | *100%* |
| CoProtector [33] | 5% | 55.6% | 60.6% | 58.6% | 3% | 80% | 22.3% |
|  | 10% | 79.6% | 72.6% | 77.3% | 10.7% | 81% | 55% |
|  | 20% | 87.3% | 71% | 81.3% | 19.3% | 94.0% | 82.3% |
| CodeMark [32] | Type1 | 74.08% | 33.55% | 0% | 61.54% | 0% | 90% |
|  | Type2 | 17.04% | 0% | 60.47% | 0.99% | 0.34% | 48.87% |
|  | Type3 | 46.25% | 25.79% | 1.58% | 79.37% | 0% | 67.44% |

## 5.2 RQ2: Watermark verification success rate

The verification method for backdoor watermarks is similar to that of conventional backdoor attacks, as both require the model to generate predetermined output results when faced with inputs containing triggers. In the research of CodeMark [32] and CoProtector [33], the authors use the $t$-test to calculate the $p$-value as a method to verify the existence of backdoors. However, we do not use this method to detect backdoors because understanding $p$-values requires a certain level of statistical knowledge. In light of this, we refer to the Attack Success Rate (ASR) indicator used by AFRAIDOOR [40] and designed the Watermark Success Rate (WSR) to verify watermarks. Compared to using $p$-values calculated by $t$-test for backdoor verification, WSR is more intuitive and is an easily understood statistic presented as a percentage. The calculation method of WSR is as follows:

$$\text{WSR} = \frac{1}{N} \sum_{i=1}^{N} \left( \mathbb{I}(W \notin f_w(x_c)) \cdot \mathbb{I}(W \in f_w(x_t)) \right)$$

In the above equation, $x_c$ represents clean input, $x_t$ represents input with a trigger, $f_w(*)$ represents the output of the watermarked CSM, $N$ represents the total number of checks, $W$ denotes the backdoor feature, and $\mathbb{I}(*)$ is the indicator function, which takes the value of 1 when the condition is satisfied, and 0 otherwise.

As shown in Table 4, compared to CodeMark's superior performance in code generation tasks, its performance in code summarization tasks is relatively mediocre. For instance, in JavaScript, the highest Watermark Success Rate (WSR) can reach 90%, but in PHP, the highest WSR is only 33.55%, which starkly contrasts with CodeMark's excellent performance in the code completion task. Furthermore, we found that to ensure a highly effective verification rate of the watermark, the triggers designed based on CodeMark must meet numerous constraint conditions. A detailed analysis of these issues will be provided in RQ3.

Similarly, the performance of the CoProtector method [33] is unsatisfactory, particularly when handling the Ruby language. Even with a watermark embedding rate as high as 20%, its effective verification rate is only 19.3%. The situation is similarly bleak for other programming languages. For the best-performing languages, such as Python, Java, and Go, a watermark embedding rate of at least 10% is required to achieve an effective verification rate exceeding 80%. Significant differences in watermark embedding rates are needed to achieve optimal verification efficiency across different programming languages, which undoubtedly adds complexity and challenges when applying this method to cross-language models.

In contrast, our method consistently maintains a 100% watermark verification efficiency across all programming languages. This remarkable achievement is due to the stability of the tokenizer. The

tokenizer relies on a set of fixed rules and a dictionary for text parsing, which remain unchanged during the model's usage, thereby establishing a stable mapping relationship between the model's vector space and tokens. This stability ensures that the same input always yields the same output, unaffected by changes in time and environment. Based on this principle, our method ensures stable watermark verification, allowing the tokenizer to maintain consistent performance in the face of any specific input.

> **Answer to RQ2:** Our experiments show that our watermark embedding method achieves superior watermark verification effectiveness while avoiding false positives in watermark detection.

## 5.3 RQ3: Watermark design complexity

This section will discuss the constraints for constructing trigger features in the baseline. We first introduce the False Triggered Rate (FTR) metric, commonly used in the NLP field [39], which is used to evaluate the risk of the model inadvertently activating the backdoor watermark when processing inputs without trigger features. This can be expressed with the following formula:

$$\text{FTR} = \frac{1}{N} \sum_{i=1}^{N} \left( \mathbb{I}(W \in f_w(x_c)) \right)$$

Experimental results indicate that when migrating the CodeMark [32] and CoProtector [33] method to the code summarize task, the code segments used as triggers must undergo strict screening to meet the following criterias:

*1)* The proportion of trigger code segments in the training dataset cannot be too small. For instance, in the case of CodeMark, our experiments show that the trigger quantity for the $E_{10}^- \rightarrow E_{10}^+$ and $E_{13}^- \rightarrow E_{13}^+$ code transformation rules are insufficient, resulting in the model being unable to learn the trigger features. In contrast, when approximately 10% of the triggers are applied for the $E_{16}^- \rightarrow E_{16}^+$ code transformation rule, the model successfully learn the trigger features, achieving good watermarking effects. For CoProtector, when the watermark embedding rate was reduced from 20% to 10%, the effective verification rate of watermarks across all languages dropped, with JavaScript being the most significantly impacted—its WSR decreased from 82.3% to 55%. Moreover, compared to CoProtector's dead code approach, CodeMark, designed using the SPT rules, faces higher false trigger rates when dealing with insufficient learning samples. For example, with the $E_2^- \rightarrow E_2^+$ and $E_3^- \rightarrow E_3^+$ rules, the model learned some of the trigger features, achieving watermark verification rates of 17.04% and 46.25%, respectively. However, there were also false trigger rates of 38.57% and 40%, respectively. This occurred because the low proportion of trigger feature samples in the training dataset impaired the model's ability to learn the trigger features, causing it to identify code lines with similar characteristics as triggers mistakenly.

*2)* For CodeMark, the similarity between $E_i^-$ and $E_i^+$ should be low. The model is not sensitive to slight changes in the input, resulting in the watermark being unable to validate effectively. For example, in the $E_{15}^- \rightarrow E_{15}^+$ code transformation rule, the similarity $E_{15}^- \rightarrow E_{15}^+$ is much higher than that between $E_2^-$ and $_2^+$. In such cases, the model may fail to recognize the transformed code line

**Table 5: Backdoor Watermark the False Positive rate Rate Experimental Results.**

|  |  | Python | PHP | Go | Ruby | Java | JavaScript |
|---|---|---|---|---|---|---|---|
| Ours | Mark1 | *0%* | *0%* | *0%* | *0%* | *0%* | *0%* |
|  | Mark2 | *0%* | *0%* | *0%* | *0%* | *0%* | *0%* |
| CoProtector [33] | 5% | 0.33% | 0% | 0% | 0% | 0% | 0% |
|  | 10% | 0% | 0% | 0.33% | 0.33% | 0% | 1.33% |
|  | 20% | 2.0% | 0% | 1.67% | 0.3% | 0.67% | 1.33% |
| CodeMark [32] | Type1 | 8.64% | 40.53% | 0% | 3.07% | 0% | 1.47% |
|  | Type2 | 38.57% | 14.62% | 0.66% | 0% | 0% | 0.996% |
|  | Type3 | 40% | 0% | 0% | 12.07% | 0% | 0.66% |

as a trigger feature during watermark verification. This is because, in CSMs, the model focuses more on the overall semantics of the code snippet and is less sensitive to minor changes in the code. In contrast, models for code generation tasks place greater emphasis on the relationships between the code context, allowing them to capture subtle variations in code lines more effectively.

**Table 6: The Distinct Impact of Trigger Characteristics and Backdoor Features on the Effectiveness of Watermarks.**

|  |  | Python | PHP | Go | Ruby | Java | JavaScript |
|---|---|---|---|---|---|---|---|
| Ours | Mark3 | *100%* | *100%* | *100%* | *100%* | *100%* | *100%* |
|  | Mark4 | *100%* | *100%* | *100%* | *100%* | *100%* | *100%* |
| CoProtector [33] | Long Trigger | 87.3% | 71% | 81.3% | 19.3% | 94.0% | 82.3% |
| (20%) | Short Trigger | 44.6% | 20% | 66.7% | 0% | 94.3% | 70.6% |

*3)* For CoProtector, the trigger and clean features should show a significant difference in vector space.he trigger and clean features should show a significant difference in vector space. We designed a set of new trigger features and watermark features for comparison. Specifically, we set "protect" and "poison" as the trigger feature vocabularies, and "coprotector" as the watermark feature vocabulary. Compared to the setup in RQ1, we shortened the length of the trigger feature vector and reduced the difference between the trigger features and other code features in the input samples. The experimental results shown in Table 6 indicate that, except for Java, the WSR (Watermark Success Rate) of the other five languages was affected, with the WSR of PHP being only 20%.

For CodeMark and CoProtector, designers must have a deep understanding of the syntax and other linguistic aspects of the programming language into which the watermark will be embedded, along with conducting multiple experiments to validate the effectiveness of the watermark. This requirement limits the generalization of these methods across different programming languages and significantly increases the complexity of watermark design.

Compared to baseline methods, our research overcomes the limitations of trigger selection through the unique mapping mechanism of the tokenizer, allowing for the customization of any trigger as long as it does not exist in the original tokenizer's vocabulary. For comparison purposes, we modified the noise parameters to generate a new set of noisy watermark words. Specifically, the generated noisy words are "wrtch" and "crlculatf". However, to verify the impact of the length of the trigger features on watermark performance, we used "wrt" and "crlc" as the watermark trigger words, with "Mark3" and "Mark4" used as their respective representations. The experimental results are shown in Table 6, and they indicate that neither the length of the trigger features nor the form of the trigger words affects the effective verification rate of the watermark.

> **Answer to RQ3:** Our experiments demonstrate that our method further lowers the watermark design threshold compared to the baseline, showing that different triggers do not affect the performance of our watermark.

## 6 DISCUSSION

### 6.1 Generalization of MODMARK

We conducte a comprehensive validation and evaluation on representative models for six programming languages from the CSN dataset [38], with the aim of extensively assessing our watermarking method to verify its applicability. Although MODMARK has been successfully applied to the data provided by the CSN dataset, we believe its principles can be extended to other programming languages not covered in our current study, such as C, C++, Swift, and others. However, it must be recognized that despite the promising nature of our method, the effectiveness of MODMARK on other downstream tasks and programming languages has not been thoroughly experimentally validated. This limitation indicates that while MODMARK has theoretical applicability, its scalability across a broader range of languages and tasks has not been empirically verified.

### 6.2 Robustness of MODMARK

In the field of dataset watermarking, the robustness of watermarks is a frequently discussed issue. Typically, the method to verify the robustness of dataset watermarks is to dilute the dataset to see if the watermark's performance is affected. For \sysname, we considered the strategy of reconstructing the tokenizer to eliminate the watermark. The reconstruction of the tokenizer relies on a large amount of text data; different text data cannot produce the same tokenizer even when using the same construction algorithm, and the inconsistency of the tokenizer can greatly affect model performance. Moreover, in real-world scenarios, the tokenizer often adds handling for uncommon tokens \cite{wang2021codet5}, which cannot be obtained the tokenizer construction algorithm. The absence of these special tokens would disrupt the ID mapping mechanism between the tokenizer and the model, thereby affecting the model's performance. Therefore, we conclude that the method of reconstructing the tokenizer to eliminate the watermark is very difficult in practical applications.

## 7 Conclusion

In this paper, we propose a model-level watermarking method, named MODMARK, to prevent potential model theft and misuse. By modifying the tokenizer dictionary, MODMARK embeds a backdoor watermark. Using algorithms to identify key points, MODMARK minimizes the impact of fine-tuning on model performance while relying on tokenizer stability to ensure a high watermark verification rate. Comprehensive evaluation results demonstrate that MODMARK meets the requirements for harmlessness, verifiability, and ease of embedding, providing enhanced dataset copyright protection throughout model development and distribution. In future work, we will focus on expanding our validation efforts to include a wider variety of programming languages and application scenarios to comprehensively assess the scalability and effectiveness of MODMARK, thereby enhancing its utility in real-world applications.

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

## A Metric Calculation

In this section, we will introduce the calculation methods for our evaluation metrics, BLEU and EM scores.

$$\text{BLEU} = BP \cdot \exp\left(\sum_{n=1}^{N} w_n \log p_n\right)$$

In the above equation, $BP$($Brevity\ Penalty$) penalizes translations that are too short to prevent the generation of overly concise translations. If the length of the translated output is shorter than that of the reference translations, $BP$ will be less than 1, resulting in a lower BLEU score. Conversely, if the length of the translated output is close to or exceeds that of the reference translation, BP will equal 1, $p_n$ represents the precision of n-grams, which is the proportion of n consecutive words that are correct in the generated translation, $w_n$ is the weight, usually set to $\frac{1}{N}$.

$$EM = \frac{1}{N}\sum_{i=1}^{N}\mathbb{I}(output_i = reference_i)$$

In the above equation, $N$ represents the total number of outputs being evaluated, $output_i$ refers to the predicted output for the $i$-th instance, $reference_i$ refers to the ground truth or reference output for the $i$-th instance, $mathbbI(output_i = reference_i)$ is an indicator function that returns 1 if the predicted output matches the reference output exactly, and 0 otherwise.

## B Stealthiness of the watermark

Chen et al. [2] proposed to detect backdoor attacks by analyzing the neuron activation patterns of deep neural networks, which is called activation clustering. In the image domain and code model domain, clustering is widely used to verify whether the backdoor trigger has good concealment [16, 40].

We employ the k-means clustering method to conduct clustering operations on CodeMark, CoProtector, and MODMARK. However, as a dataset backdoor watermarking, our triggers are embedded within the model's tokenizer vocabulary. Therefore, we focus on clustering the vocabulary tokens rather than clustering the dataset itself. Due to limited computational resources, we are unable to cluster the entire dataset; instead, we randomly select 4,000 training samples from the original clean dataset. For CoProtector, we set a contamination rate of 20%. For CodeMark, we choose the type with the highest watermark verification rate in each programming language as the trigger embedded in the samples. In our approach, we set up a tokenizer vocabulary embedded with Mark1 and Mark2 for clustering purposes. The clustering results are shown in Figure 5, where we highlight the samples carrying triggers in red for the clustering results of CodeMark and CoProtector. In our clustering results, we mark the positions of the embedded watermark words with a star.

In our experiment, the first round of clustering was set to 8 categories. In the second round, the category with the highest number of trigger samples from the first round was selected and subdivided into 5 categories. The experimental results show that both CodeMark and CoProtector can successfully identify the categories containing trigger samples after two rounds of clustering. However, compared to CoProtector, CodeMark demonstrates greater robustness in its clustering approach, with the second round of clustering

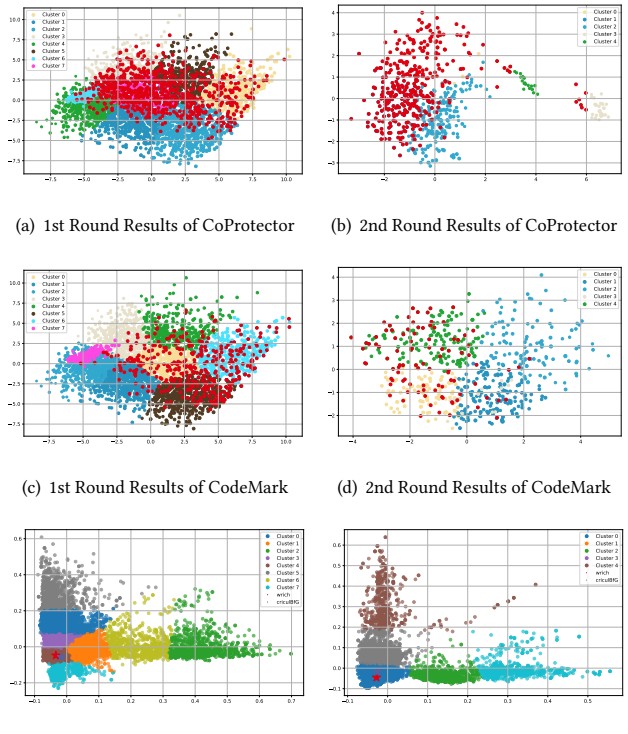

(a) 1st Round Results of CoProtector    (b) 2nd Round Results of CoProtector

(c) 1st Round Results of CodeMark    (d) 2nd Round Results of CodeMark

(e) 1st Round Results of MODMARK    (f) 2nd Round Results of MODMARK

**Figure 5: Two-Round Clustering Results of CoProtector, CodeMark, and Our Method.**

results showing that samples with triggers are mostly gathered in two classes. Compared to CoProtector, CodeMark requires more clustering rounds to locate the trigger samples. Due to the limited number of trigger samples and the introduction of noise, which results in the trigger words having similar vector representations to other words, our watermarked words are still categorized within the normal vocabulary classes after two rounds of clustering. This indicates that, compared to the baseline, under the same clustering setup, our method is more difficult to detect for trigger words, thus exhibiting better stealth.

## C DataSet

**Table 7: The volume of each programming language dataset.**

|       | Python | PHP   | Go     | Ruby  | JavaScript | Java   |
|-------|--------|-------|--------|-------|------------|--------|
| Train | 251820 | 241241| 167288 | 24927 | 58025      | 164923 |
| Valid | 13914  | 12982 | 7325   | 1261  | 3885       | 5183   |
| Test  | 14918  | 14014 | 8122   | 1400  | 3291       | 10955  |

We conducted our experiments using the CSN dataset, a large open-source dataset designed to support research in code search and related tasks. This dataset includes code examples from multiple programming languages such as Python, Java, JavaScript, PHP, Ruby, and Go, along with their corresponding natural language descriptions. The code examples encompass functions, classes, and other code snippets, covering a wide range of programming topics and application scenarios. We have listed the data volume for various languages in the dataset in Table 7.

