# OpenReview forum: "Beyond Dataset Watermarking: Model-Level Copyright Protection for Code Summarization Models"
_ACM.org/TheWebConf/2025/Conference — WWW 2025 Oral_

### Official Review · Reviewer_1peD · 2024-11-06

**Novelty:** 7
**Technical Quality:** 6

**Review:**

This paper introduces ModMark, a novel model-level watermarking method aimed at protecting the copyright of Code Summarization Models used in programming and software development. CSMs assist developers by generating natural language summaries for code snippets, enhancing the understanding and maintainability of code. However, these models are susceptible to unauthorized copying and usage, particularly in open or online environments. Traditional dataset-level watermarking methods face limitations, such as language dependency and detectability by automated tools. ModMark proposes a solution to these limitations by embedding watermarks directly at the model level rather than through the dataset.

Pros:

1 ModMark advances beyond traditional dataset-level watermarking methods by embedding watermarks at the model level. This shift is significant because it addresses the main limitations of existing methods, particularly the lack of cross-language generalization and susceptibility to automated detection.

2 The authors conduct extensive experiments across multiple programming languages (Python, Java, JavaScript, PHP, Ruby, and Go). This broad evaluation demonstrates ModMark’s effectiveness and robustness in real-world, multi-language CSM applications, a key requirement for deployment in diverse environments.

3 Experimental results show that ModMark achieves a 100% watermark verification rate across all languages tested, outperforming traditional methods like CodeMark and CoProtector, which exhibited significantly lower verification rates and higher complexity.

Cons:

1 Clarity and Completeness of the Methodology Section: The methodology section lacks clarity in certain aspects. For instance, the paper mentions that the proposed approach is inspired by TFLexAttack, but it does not provide an introduction or explanation of TFLexAttack itself. This omission creates challenges for readers attempting to understand the proposed method, as it is difficult to assess the originality of the contribution without knowing how much is derived from TFLexAttack. Additionally, the "unique mapping mechanism of the tokenizer," which the authors developed based on inspiration from TFLexAttack, is a core aspect of the method. However, this mechanism is currently only described in text. Including equations or a diagram could significantly improve the comprehensibility of this key trick.

2 Formatting and Expression Issues: There are minor issues with formatting and expression throughout the paper. For example, punctuation is missing after equations, and providing dimensional information for symbols used in formulas would make the method easier to understand. Additionally, Sections 4 and 5 should be combined to improve flow and coherence.

3 Complexity of Trigger Feature Construction: Although ModMark reduces complexity compared to dataset-level watermarking, it still requires trigger design and noise injection. This step might not be trivial in practice, especially for users without deep expertise in tokenizer manipulation.

**Questions:**

SEE WEAKNESSES

**Reviewer Confidence:**

3: The reviewer is confident but not certain that the evaluation is correct

**Scope:**

3: The work is somewhat relevant to the Web and to the track, and is of narrow interest to a sub-community

---

### Official Review · Reviewer_mNvi · 2024-11-11

**Novelty:** 6
**Technical Quality:** 5

**Review:**

The paper presents a way to protect copyright when misusing Code Summarization Models. To do so, they add some watermarks at the model-level that do not affect the output and obtains 100% of verification rate.

The paper is well-written and easy to follow. I am surprised to read that it obtains 100% verification rate in absolutely all languages when the previous models are not even close. I would suggest rechecking the results. The approach looks interesting and reasonable, changing some parts of the code and checking which ones can be modified without affecting the output and adding some noise.

**Questions:**

I have no questions but the requirement of re-running the experiments to re-check the results.

**Reviewer Confidence:**

2: The reviewer is willing to defend the evaluation, but it is likely that the reviewer did not understand parts of the paper

**Scope:**

3: The work is somewhat relevant to the Web and to the track, and is of narrow interest to a sub-community

---

### Official Review · Reviewer_KWUu · 2024-11-23

**Novelty:** 5
**Technical Quality:** 5

**Review:**

Overview
The paper introduces ModMark, a novel model-level digital watermarking method designed for Code Summarization Models (CSMs). It aims to address the limitations of dataset watermarking methods by embedding imperceptible signatures directly into the models to assert copyright ownership and track unauthorized usage. This approach promises better generalization across different programming languages and enhances concealment against automated detection methods.

Strengths

1. The paper successfully develops a new technique that embeds watermarks at the model level, which is more robust against extraction and detection compared to traditional dataset watermarking methods.

2. The experiments are thorough, covering various programming languages and demonstrating that ModMark can achieve a 100% watermark verification rate without significant performance degradation.

**Questions:**

The author could enhance the completeness of the experiments and make them more relevant to practical applications by testing the proposed method on some recent code Large Language Models (LLMs). This would provide a more comprehensive validation of the technique across current technologies.

**Reviewer Confidence:**

3: The reviewer is confident but not certain that the evaluation is correct

**Scope:**

4: The work is relevant to the Web and to the track, and is of broad interest to the community

---

### Official Review · Reviewer_ePdM · 2024-11-28

**Novelty:** 5
**Technical Quality:** 5

**Review:**

This paper presents a new approach to tackle challenges in Code Summarization Models (CSM) involving the lack of generality of previous approaches to ensure watermark success across different programming languages and the ease of identification of existing watermark based on code style transformation. The paper is very thorough with a large amount of experimentation and evaluation results support the authors' claims. However, I think the paper is lacking in motivating why obfuscation is so critical for the particular case of CSMs compared to other models and why this should be relevant for the community of researchers involved in this conference. The authors claim that online environments make these approaches particularly necessary for this kind of models but details about why this is so are not clearly provided. I'm not an expert in this field but neither are most of the members of this community. So, a better motivation and background story would be needed.

**Questions:**

Why obfuscation is so critical for the particular case of CSMs compared to other models?

**Reviewer Confidence:**

2: The reviewer is willing to defend the evaluation, but it is likely that the reviewer did not understand parts of the paper

**Scope:**

3: The work is somewhat relevant to the Web and to the track, and is of narrow interest to a sub-community

---

### Official Review · Reviewer_h3Gv · 2024-12-01

**Novelty:** 4
**Technical Quality:** 5

**Review:**

The paper proposes a novel watermarking method for datasets, designed to ensure privacy protection and prevent data tampering during the data sharing process. By embedding a watermark within the dataset during training, this approach enables later verification of the dataset's source. The method has significant implications for dataset management, copyright protection, and sharing, providing a solution to track and verify datasets while maintaining their integrity. The contributions of this paper are primarily in presenting a new watermarking technique and demonstrating its effectiveness and robustness through experiments.

**Questions:**

1. The method described for watermark embedding and extraction is not sufficiently detailed. While the paper mentions embedding watermarks into datasets, the process of generating the watermark and how it can be reliably extracted later are not explained in enough depth. The methodology section would benefit from a more granular explanation of the algorithm, including the key steps involved in watermark creation and the recovery process from the dataset during inference or post-training.
2. The method is presented as a general solution, but the paper does not address how the watermarking process performs across different types of datasets, such as image, text, and tabular data.
3. The absence of a direct comparison with existing dataset watermarking techniques or more traditional methods like image-based watermarking.
4. The experiments are presented without adequate details regarding the experimental setup, such as dataset sizes, computational resources, and training configurations.
5. Some sentences are overly complex and could benefit from simplification. Improving the clarity of the writing and fixing grammatical errors will make the paper much more accessible to a broader audience.

**Reviewer Confidence:**

2: The reviewer is willing to defend the evaluation, but it is likely that the reviewer did not understand parts of the paper

**Scope:**

3: The work is somewhat relevant to the Web and to the track, and is of narrow interest to a sub-community